# Efficient Visualization of Implicit Neural Representations via Weight Matrix Analysis

## Abstract

An implicit neural representation (INR) is a neural network that approximates a function over space and possibly time. Memory-intensive visualization tasks, including modern 4D CT scanning methods, represent data natively as INRs. While such INRs are prized for being more memory-efficient than traditional data on a lattice, discretization to a regular grid is still required for many visualization tasks. We present an algorithm to store high-resolution voxel data only for regions with significant detail, reducing memory requirements. To identify these high-detail areas, we use an interpolative decomposition pruning method on the weight matrices of the INR. The information from pruning is used to guide adaptive mesh refinement, allowing automatic mesh generation, tailored to the underlying resolution of the function. From a pre-trained INR with no access to its training data, we produce a variable resolution visualization with significant memory savings.

## 1 Introduction and motivation

Implicit neural representations (INRs) have gained traction in recent years for their ability to represent spatial and time-varying spatial data efficiently. While INRs are best known for their fast and accurate visualization applications, these methods only apply to specific neural graphics primitives—such as signed distance functions—and require training routines and data structures—such as hashing techniques—to realize interactive visualization. For INRs encoding data not derived from graphics primitives, the recourse for visual analysis is to discretize the INR to a uniform grid, thereby enabling traditional techniques, but eliminating any computational savings afforded by the INR encoding. This presents an open challenge to communities using INRs in new contexts: given a pre-trained INR, how can the information encoded be visualized efficiently?

The need for efficient visualization of INR data is evidenced by emergent "dynamic micro-CT" technology for additive manufacturing. Recently developed methodologies are capable of storing time-varying volumetric data of materials undergoing physical changes as an INR with $(x, y, z, t)$ inputs. In one example, the size of an INR checkpoint file is on the order a few megabytes, but the potential resolution of the time-varying volume is $1024 \times 1024 \times 1024 \times 700$, roughly 3.6 *terabytes* worth of data in a uniform discretization, well beyond the capabilities of common visualization software. By visual inspection of time slices, many regions of the domain are of low variation while some regions require maximum resolution for subject matter expert evaluation. Hence, an approach to adaptively sample the INR in a way that preserves fine-grained details of the function is of real interest to practitioners with immediate benefits to dynamic micro-CT technology.

In this paper, we present an algorithm that visualizes a pre-trained INR on an adaptive mesh, achieving accuracy comparable to a uniform mesh while using less memory. The algorithm begins with a coarse uniform mesh of the domain and iteratively refines elements in which the INR is expected to encode finer-scale information. We assume knowledge of the INR architecture, as would be encoded in a standard checkpoint file, but we do not assume access to any training data; the algorithm determines where to refine based solely on the weight matrices of the INR. The refinement decision for a given element is based on a "pruning" method applied on the INR, restricted to the element's domain. Elements for which significant pruning is possible with small loss in accuracy are presumed to have low-rank representations and therefore deemed sufficiently refined. Conversely, elements for which significant pruning is not possible, or for which pruning causes significant information loss, are flagged for refinement.

## 2 BACKGROUND AND LITERATURE COMPARISON

### 2.1 IMPLICIT NEURAL REPRESENTATIONS

An implicit neural representation (INR) is a type of neural network that approximates a scalar- or vector-valued field with inputs representing physical space or spacetime coordinates. The original use of INRs in the context of visualization was to efficiently store an implicit representation of an image (Sitzmann et al., 2020), but interest in the technique quickly grew to include volumetric visualizations as well (Mildenhall et al., 2021). The output of the popular physically-informed neural network (PINN) technique for approximating solutions to partial differential equations is a coordinate-valued, multi-layer perceptron (typically), and hence could also be called an INR (Karniadakis et al., 2021).

The appeal of INRs over traditional discretization is, to quote Sitzmann et al. (2020), the network's "ability to model fine detail that is not limited by the grid resolution but by the capacity of the underlying network architecture." Only the weights and biases of the INR need to be stored in order to recover the value of the field at the highest level of detail anywhere in the represented domain. Accordingly, the INR data structure takes up orders of magnitude less storage than an equivalent standard representation. Still, the savings in data storage come with a tradeoff: evaluating the INR can only be done "pointwise", meaning discretization and interpolation over a fixed grid of some type is required to employ standard visualization software for all but very specific types of INR data.

### 2.2 VISUALIZATION AND DISCRETIZATION

While our work is related to both visualization using INRs and traditional data discretization methods, neither of the associated research communities offers a solution to the problem we are addressing. Much of the visualization work on INRs focuses on methods to train INRs more efficiently, such as ACORN (Martel et al., 2021), scene representation networks (Wurster et al., 2023), and Instant-NGP (Wurster et al., 2023). None of these works, however, addresses the question of how to process, analyze, or efficiently visualize a pre-trained INR. A separate body of work looks at efficient management and visualization of data stored on adaptive meshes, such as multi-functional approximation (Peterka et al., 2023), CPU ray tracing (Wang et al., 2020), and p4est (Burstedde et al., 2011). These works presume data is provided on an adaptive mesh as input to their use cases, rather than as a pre-trained INR.

We treat INRs as a native data format, akin to a compressed version of a much larger dataset. The input to our method is a user-provided INR, with no access to the training data. As output, we produce an adaptive mesh on which the INR has been sampled at vertices, allowing subsequent visualization and analysis via established techniques. To the best of our knowledge, there is no prior work considering this problem, other than sampling to a uniform grid.

### 2.3 PRUNING VIA INTERPOLATIVE DECOMPOSITION OF WEIGHT MATRICES

"Pruning" refers to the process of selectively removing weights and biases from a trained neural network in a way that preserves its mapping from inputs to outputs; see, e.g. (Li et al., 2016; Lee et al., 2018; Liu et al., 2018; Liebenwein et al., 2019; Mussay et al., 2019). We use the pruning method of Chee et al. (2022), which merges neurons in each layer whose contributions to the output are close to a linear combination. The method for detection of such neurons employs a structured low-rank approximation called an "interpolative decomposition" (ID). We selected this pruning method due to its theoretical guarantees, ease of implementation, and few number of hyperparameters.

We fix notation before describing the ID pruning method. In this work, we only consider INRs that consist of fully-connected linear layers. Hence, each layer takes as input $x \in \mathbb{R}^n$, provides an output $y \in \mathbb{R}^m$ and has corresponding weight matrix $W \in \mathbb{R}^{m \times n}$ and bias vector $b \in \mathbb{R}^m$. We treat the inputs $x$ and outputs $y$ as row vectors and assume that the output of the layer is computed as

$$y = g(xW^T + b),$$

where $g$ is the activation function used for the layer. If a collection of $\ell$ inputs is provided, we still use $x$ to denote the $\ell \times n$ matrix of inputs yielding an output $y \in \mathbb{R}^\ell$.

An ID of $W$ is a decomposition of the form $W \approx W_{:,\mathcal{I}}T$, where $\mathcal{I} \subseteq \{1, 2, ..., m\}$, $|\mathcal{I}| = k$, and $T \in \mathbb{R}^{k \times n}$ is called the "interpolation matrix." For the ease of exposition, suppose the neural network has a single hidden layer with output layer weight matrix $U$ and output layer bias vector $c$. Then, the output of the network with input $x$ is $NN(x) := Z(x)U^T + c$, where $Z(x) = g(xW^T + b)$ is the output of the hidden layer.

Let $Z(x) \approx Z_{:,\mathcal{I}}T$ be an ID of $Z(x)$. We then have

$$Z(x) \approx Z_{:,\mathcal{I}}T$$
$$= g(xW^T + b)_{:,\mathcal{I}}T$$
$$= g(x(W_{\mathcal{I},:})^T + b_{\mathcal{I}})T.$$

Thus, the output of the full network with pruned hidden layer is

$$NN(x) = g(x(W_{\mathcal{I},:})^T + b_{\mathcal{I}})TU^T + c$$
$$= g(x\bar{W}^T + \bar{b})\bar{U}^T + c,$$

where we define $\bar{W} := W_{\mathcal{I},:}$, $\bar{b} := b_{\mathcal{I}}$, and $\bar{U} := UT^T$ to be the new weights and biases of the pruned network. Thus, pruning a layer not only affects the weights and bias of that layer, but also the weights of the following layer. The result of pruning a given layer to rank $k$ is that the resulting pruned layer has $k$ neurons. The following layer's weights are updated to accept the new, smaller number of inputs coming from the previous layer.

Given $\varepsilon > 0$, the goal of pruning is to find $\mathcal{I}$ and $T$ such that $\|W - W_{:,\mathcal{I}}T\|_2 \leq \varepsilon\|W\|_2$, with $|\mathcal{I}|$ as small as possible. We use the rank-revealing QR factorization approach from Chee et al. (2022) to carry this out. For neural networks with more than one hidden layer, IDs for each layer's weight matrix can be computed in parallel, but the final weights of the pruned network must be determined sequentially from the ID of the first layer forward.

## 3 ALGORITHM

The goal of our algorithm is to visualize an INR without computing and storing the complete fine-scale voxel data necessary to see high resolution details. By finding a suitable adaptive mesh for visualization, we avoid expending compute time and memory evaluating regions of the INR domain that are less *detailed*. We use the word *detailed* to describe a region of the domain where the function has large variation, which would be harder to fit accurately with a neural network of few parameters.

Since we only presume access to the weights and biases of the INR, we cannot easily determine regions of high variation. Instead, we rely on the hypothesis that the less detailed a function is on a region of the domain, the smaller an INR needs to be to accurately describe the function in that region. If this hypothesis holds true, a *less-detailed* region of the INR should admit more pruning with minimal loss in accuracy over that region . Furthermore, we observed that an INR evaluated on small subsets of a domain can generally be pruned much more than for the whole domain. Thus, if a region of the domain is not very prunable, then by splitting it into more, smaller regions, the sub-regions are more likely to be prunable. This also makes sense because we are asking the INR to describe less information if we restrict it to a smaller domain, so we expect to be able to use a smaller network to do so. This is the motivation for our algorithm.

To decide which regions to check for prunability (i.e., the proportion of neurons that can be pruned while maintaining an accuracy threshold), we start with an initial mesh on the INR's domain and use adaptive mesh refinement (AMR) to subdivide some elements into smaller ones. We keep refining elements until the proportion of neurons left after pruning is below a threshold, which we denote $P$, and the relative error of the pruned INR is less than a desired value, which we denote $T$. We check both of these thresholds to ensure that a small network can accurately represent the INR on that domain.

Let `prune(INR, domain, ` $\varepsilon$ `, ID_samples)` be a function that prunes an INR using an ID method. The "domain" input is the region of the domain considered for pruing, $\varepsilon$ is the maximum relative error we allow for the ID used in pruning, and ID_samples is the number of samples we use to compute the ID. See Table 3 for more information about these hyperparameters. Furthermore, let error_samples denote the number of samples used to compute the error estimate that we compare

| hyperparameter | description | heuristic |
|---|---|---|
| ID_samples | Number of samples of a given domain to take when computing the ID | This can be set to the width of the INR layers |
| $\varepsilon$ | The relative error achieved by the ID; this affects how many neurons get pruned | $10^{-3}$ |

Table 1: Descriptions for hyperparameters used in ID pruning.

| hyperparameter | description | heuristic |
|---|---|---|
| error_samples | Number of samples of a given domain to take when computing the approximate error of the pruned INR | 32 |
| T | The relative error below which a pruned INR must be to not refine the corresponding element | The main hyperparameter to decide how high of resolution you want to see |
| P | The proportion of neurons relative to the full INR that a pruned INR must have less than to not refine the corresponding element | 0.15 |
| max_it | The maximum number of iterations to refine for | Set based on limits of your machine |

Table 2: Descriptions for hyperparameters used in Pruning AMR.

against the error threshold $T$ to decide if we need to refine. We refine for up to max_it iterations. A second table summarizing all of the AMR hyperparameters along with some helpful heuristics is shown in Table 3.

For notational simplicity, assume that Mesh is a class that has a member for each element in the corresponding mesh. Each element has an attribute for its domain and another to specify whether it is done being refined or not; there is also a function random(n, domain) that can sample $n$ points from a uniform distribution on a domain. Given this notation, our algorithm for performing refinement using a pruning-based error estimate is given in Algorithm 1.

---
**Algorithm 1:** Algorithm Pruning AMR: using adaptive mesh refinement to find a memory-efficient visualization of an INR.

---
**input** : INR, inital mesh $M$, error threshold $T$, proportion threshold $P$, interpolative
        decomposition error limit $\varepsilon$, maximum number of iterations max_it, number of samples
        for error check error_samples, number of samples ot use for ID ID_samples.

**for** $it = 1$ **to** *max_it* **do**
    **for** *each element $E$ in $M$ with $M.E.done\_refining$ == False* **do**
        INR_pruned = prune(INR, $E$, $\varepsilon$, ID_samples) // prune INR on element E;
        proportion = INR_pruned.num_neuron / INR.num_neurons // compute proportion of
         neurons remaining after pruning;
        // compute error of pruned INR on element E;
        $X$ = random(error_samples, $M.E$.domain) // sample random points;
        error = mean(|INR($X$) - INR_pruned($X$)| / |INR($X$)|) // compute mean relative error;
        // Refine all elements that don't meet error or proportion threshold;
        **if** *error $> T$ or proportion $> P$* **then**
            $M.E$.refine();

**output:** Refined mesh $M$

---

# 4 RESULTS

## 4.1 2D VALIDATION EXAMPLE: ANALYTICAL OSCILLATION AT A CORNER

We verify and validate Algorithm 1 by testing on an INR fit to a benchmark function from the adaptive mesh refinement community (Mitchell, 2013, Section 2.8). Drawing samples of the function $f(r) := \sin(1/(\alpha + r))$ on $[0, 1]^2$, where $r$ is the radius and $\alpha = 1/50$, we train a simple ReLU feed-forward network with 4 layers of width 32. Sampled to the vertices of a regular mesh of $512 \times 512$ square elements, and visualized with bilinear interpolation, it is evident that the oscillations of the function have been captured to a fine resolution by the trained INR; see Figure 2 (left column). We use the open source software MFEM to manage the adaptive meshing and GLVis to generate the 2D figures.

We consider two alternatives to Algorithm 1 for comparison: `Uniform` refinement and `Basic` adaptive refinement. The `Uniform` method carries out refinement on every element until a maximum number of iterations are reached. The `Basic` method takes in an integer $error\_samples$ and a threshold $\tau$. The inner loop of Algorithm 1 is replaced by drawing $error\_samples$ random points in $E$, computing the mean relative error of the INR with respect to the bilinear interpolant on $E$ at those points, and refining if the relative error is larger than $\tau$. Note that the relative error computed in `Basic` is distinct from the relative error of the pruned INR computed in Algorithm 1, hence, $\tau$ should not be equated with $T$.

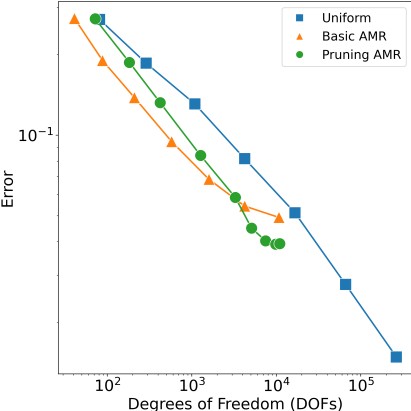

Figure 1: Total error versus number of degrees of freedom plots are shown for `Uniform`, and the best-tuned instances of `Basic` ($\tau = 0.1$) and `Pruning` ($T = 0.1$, $P = 0.09$, and $\varepsilon = 10^{-3}$). The `Pruning` method—i.e. Algorithm 1—drives down error at a faster rate than the `Uniform` approach and terminates with a lower error for an equivalent number of DOFs than either `Basic` or `Uniform`.

To assess the effectiveness of a refinement method quantitatively, we record the number of degrees of freedom (DOFs)—equivalently, the number of vertices in the mesh—and an approximation of the total error at each iteration. The total error at a given iteration is approximated as follows: First we sample a large number of points uniformly randomly across the entire domain. For this example, we used 262,144 points. At each point, we compute the value of the true INR and the bilinear interpolant of the mesh element containing that point, using the true INR values at the element corners. The root mean squared error across all sample points is then recorded as the total error. Plotting error versus DOFs is standard practice in analysis of adaptive mesh refinement schemes.

We carried out experiments to study the effect of the key parameters for Algorithm 1 and `Basic`, namely, $P$, $T$, and $\tau$. The goal was to find parameters that minimize both total error and degrees of freedom at the termination of the algorithm. At a high level, the findings are consistent with what we expected. If $P$, $T$ or $\tau$ are too low , too many elements are refined and the result is similar to that of `Uniform`. If $P$ or $\tau$ is too high, too few elements are refined and Algorithm 1 stops after a few iterations. For the 2D example with a maximum of 9 iterations and a dof threshold of 10,000,

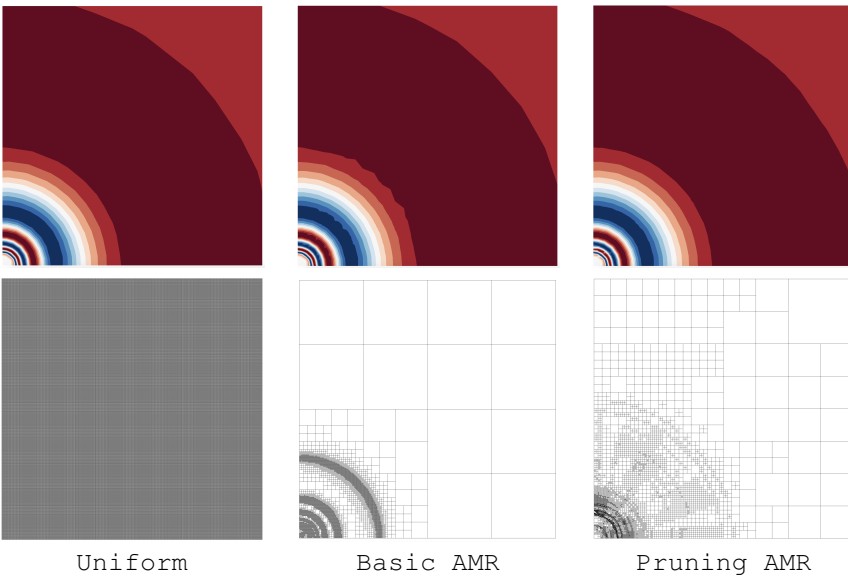

Uniform      Basic AMR      Pruning AMR

Figure 2: We compare three approaches to mesh refinement of the 2D benchmark example INR. The bottom row shows the mesh at the final state of the refinement method. The top row shows a bilinear interpolant of the INR data evaluated at vertices of the mesh. Treating `Uniform` as "ground truth," observe that `Basic AMR` has multiple level sets with inaccurate variations, whereas `Pruning AMR` is visibly more similar.

we found that for pruning we needed $P > 0.05$ and $T >$ 1e-5, while for the basic method we needed 1e-3$< \tau <$0.2. These choices of parameters are specific to the 2D example.

The best results for both Algorithm 1 and `Basic` are shown in Figures 1 and 2 . The `Uniform` method drives error down linearly (in log scale) with respect to DOFs, as is expected. The `Basic` method (with optimal parameters) makes fewer refinements than `Uniform` in the first iteration, but drives down error at a similar rate to `Uniform`, until eventually leveling out. The `Pruning` method (with optimal parameters)—i.e. Algorithm 1—refines nearly all elements in the first iteration, but then drives down error at a *faster* rate than `Uniform`, ultimately terminating at a lower error but equivalent DOF count as the `Basic` method. Furthermore, we show in Figure 2 that the final mesh produced by `Pruning` produces a qualitatively more accurate approximation to the INR than the final mesh produced by `Basic`. We contend this validates the effectiveness of Algorithm 1 as a means for adaptive mesh refinement as the need to tune parameters is a challenge affecting all adaptive refinement schemes.

## 4.2 EXAMPLE 1: SIMULATED DYNAMIC CT INR

We now consider an INR from a simulated CT scan of a 3D object being compressed in time. The object is a cube with a cylindrical hole missing from its center. At time $t = -1$ the cube is uncompressed, but as time passes the cube is compressed on four sides by rectangular prisms. See the leftmost image in Figure 5 for an overhead view. More information about the pre-trained INR can be found in Mohan et al. (2024). The architecture of the INR consists of a Gaussian random Fourier feature encoding layer (see Tancik et al. (2020)), five fully-connected layers, each with a width of 256 neurons, swish activation functions, and a linear output layer with scalar output. The inputs to the INR are $x, y, z, t$, each in the range $[-1, 1]$.

We applied Algorithm 1 ("Pruning") to the simulated CT INR and compared it to `Uniform` refinement and `Basic` AMR, as described in Section 4.1. All results for this example use the hyperparameters: $T = \tau = 0.0001$, $P = 0.075$, $\varepsilon = 0.001$, max_it = 5, and ID_samples = 256. We use error_samples = 32 for `Pruning` and error_samples = 256 for `Basic` AMR. We found these hyperparameters empirically, by keeping error_samples and $\varepsilon$ fixed and varying the accuracy

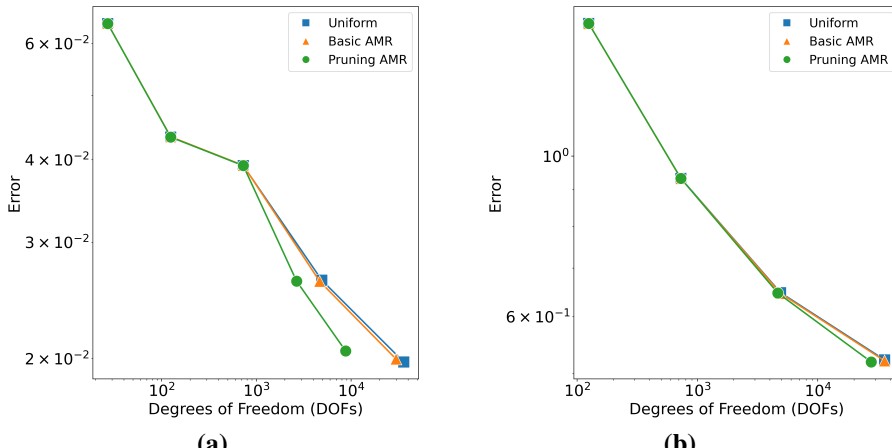

**(a)**    **(b)**

Figure 3: **(a) Simulated CT-INR example.** Total error versus number of degrees of freedom are shown for `Uniform`, `Basic` ($\tau = 10^{-4}$), and `Pruning` ($T = 10^{-4}$, $P = 0.075$, $\varepsilon = 10^{-3}$) refinement visualization of a simulated CT scan of a 3D object being compressed over time. `Pruning` achieves the same accuracy as the other two methods with significantly fewer DOFs. The gap in DOFs increases with each iteration. **(b) Experimental CT-INR example.** Total error versus number of degrees of freedom are shown for `Uniform`, `Basic` ($\tau = 10^{-3}$), and `Pruning` ($T = 10^{-3}$, $P = 0.1$, $\varepsilon = 10^{-2}$) refinement visualization of a real CT scan of a log pile. All three refinement techniques perform close to uniform refinement until the last iteration, when `Pruning` does marginally better. This example is highly-detailed and needs more iterations to show significant benefit from adaptive refinement.

thresholds ($T, \tau, P$) to target maximal accuracy within five iterations. We use $1048576$ randomly sampled points to compute the root mean squared error for all methods.

The error and DOFs for each method across five iterations are shown in Figure 3a. We require all methods to perform three uniform refinements first, since we start with a single element mesh. After these uniform refinements, we see that the `Pruning` AMR curve achieves lower DOFs for a similar level of error to both `Basic` and `Uniform`. This difference is reaffirmed in Figure 4, which shows slices of the simulated CT INR visualization for each of the three refinement methods. The top row shows slices for $x = 0$; the bottom row shows slices for $y = 0$. Both are taken at the final time, $t = 1$. For each row, the visualizations from each method appear similar. However, `Pruning` uses fewer elements (and thus, DOFs) than either `Basic` or `Uniform`. `Pruning` also seems to do a better job than `Basic` at deciding where extra elements are required. We also expect that the DOFs savings would only further improve with more iterations.

To demonstrate the utility of our algorithm in 4D, we also show the `Pruning` AMR meshes for three time slices (with $y = 0$) in Figure 5. Note that the algorithm chooses a different mesh for each time slice because the object is changing in time, even though the slices are all taken at $y = 0$.

### 4.3    EXAMPLE 2: EXPERIMENTAL DYNAMIC CT INR

Finally, we consider an INR trained on CT scans from a physical experiment. This example is much more detailed than the one in the previous section and features noise in the region surrounding the object of interest. Hence, there are fewer low-detail regions in the INR's domain.

The object of interest in this CT scan is a "log pile," which consists of many layers of strands, or "logs." Each layer has many parallel logs. The layers are rotated 90 degrees relative to each other, so that the logs in one layer are perpendicular to all of the logs in an adjacent layer. The INR used for this example has the same architecture and domain as the INR in Section 4.2. For more information about the experimental set-up and architecture, see Mohan et al. (2024).

We applied our `Pruning` AMR algorithm to the CT INR and compared it to `Uniform` refinement and `Basic` AMR, as described in Section 4.1. The results are shown in Figure Fig. 3b. For all log

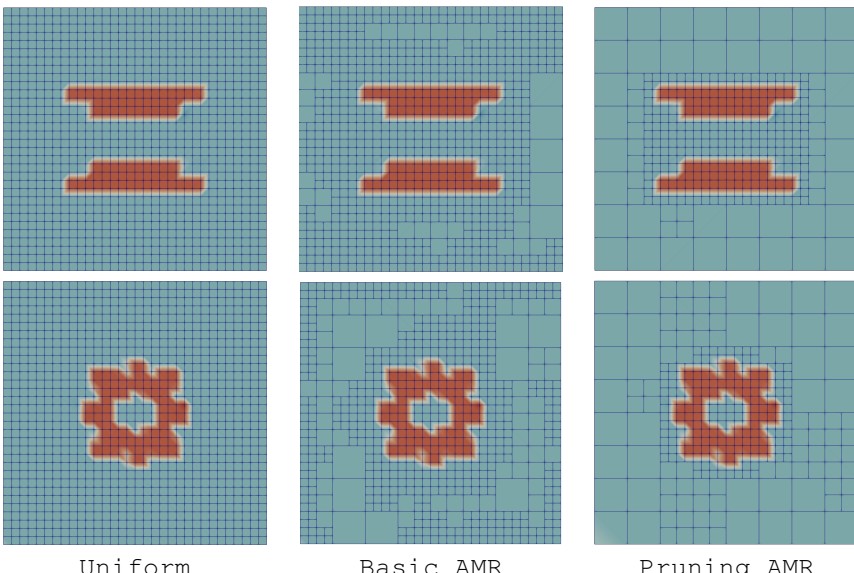

Uniform        Basic AMR        Pruning AMR

Figure 4: Comparison between meshes created by `Uniform`, `Basic` ($\tau = 10^{-4}$), and `Pruning` ($T = 10^{-4}$, $P = 0.075$, $\varepsilon = 10^{-3}$) refinement for the simulated CT INR. Top row: x-slice. Bottom row: y-slice. Each figure shows the result of five iterations of refinement. For each row, the images are visually similar, but the `Pruning` algorithm uses fewer elements than the other two methods.

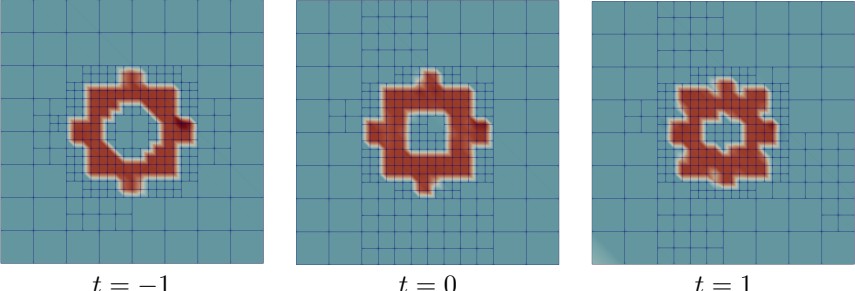

$t = -1$        $t = 0$        $t = 1$

Figure 5: Multiple time slices of simulated CT INR visualized using `Pruning` ($T = 10^{-4}$, $P = 0.075$, $\varepsilon = 10^{-2}$) AMR. Notice that the mesh changes with time as the object changes shape.

pile results, we use the hyperparameters: $T = \tau = 0.001$, $P = 0.1$, $\varepsilon = 0.01$, max_it = 5, and ID_samples = 256. We use error_samples = 32 for `Pruning` AMR and error_samples = 256 for `Basic` AMR. We use 1048576 randomly sampled points to compute the root mean squared error for all methods.

The error and DOFs for each of the three algorithms across iterations 2-5 are shown in Figure 3b. Unlike in Figure 3a, `Pruning` only does marginally better than `Basic` and `Uniform`. We believe this reflects the sparsity of low-detail regions in the dataset on which the INR was trained. Thus, both `Pruning` and `Basic` require many more iterations to get to a small enough scale to take advantage of variable detail across the domain.

Still, minor differences become apparent in the fifth iteration. For instance, consider Figure 6, which shows the log pile visualization sliced in the $x$ and $z$ direction at $t = 1$ for each of the three algorithms. On the top row ($x = 0$) we see that the `Pruning` mesh saves some DOFs in the red region of the figure where there is less variation. In the bottom row ($z = 0$), `Pruning` also saves some DOFs in the blue regions around the circular log pile. At iteration 5, these savings are minimal compared to the savings observed in the simulated CT data from Section 4.2. Thus, from this example, we confirm that AMR is only useful for INR visualization if detail is required at a scale for which there are some low-detail regions. As with the simulated data, we expect the DOFs

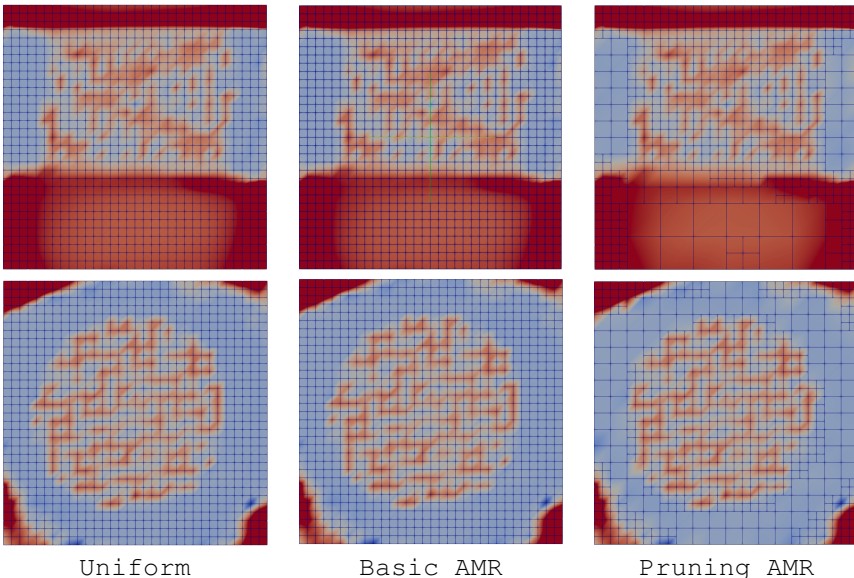

Uniform          Basic AMR          Pruning AMR

Figure 6: Comparison between meshes created by `Uniform`, `Basic` ($\tau = 10^{-3}$), and `Pruning` ($T = 10^{-3}$, $P = 0.1$, $\varepsilon = 10^{-2}$) refinement on log pile CT scan. Top row: x-slice, bottom row: z-slice. Each figure is after five iterations of refinement. For each row, the images look similar but the `Pruning` algorithm uses fewer elements in the less-detailed lower red and circular blue regions, for each row respectively.

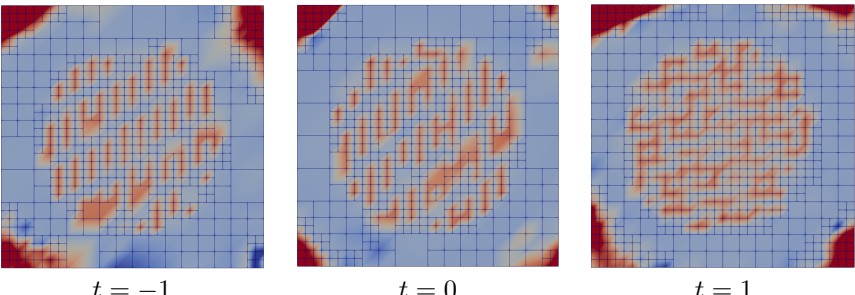

$t = -1$          $t = 0$          $t = 1$

Figure 7: Multiple time slices of the log pile CT INR visualized using `Pruning` ($T = 10^{-3}$, $P = 0.1$, $\varepsilon = 10^{-2}$) AMR. From left to right, $t = -1$, $t = 0$, $t = 1$. Notice that the mesh changes with time as the object changes shape.

savings will further improve with more iterations, however, due to computation time constraints, we were not able to investigate this further.

Finally, we demonstrate that the mesh changes with time for the experimental INR data. Figure 7 shows slices of the log pile at $z = 0$ for three different times. The mesh adapts to the shape of the object as it deforms in time.

## 5   CONCLUSION

In this paper we presented an algorithm for finding a variable-resolution visualization of pre-trained implicit neural representations (INRs) with significant memory savings over existing methods. The algorithm uses neural network pruning to determine which regions of the INR's domain require higher resolution, then uses adaptive mesh refinement to split up the domain into regions of higher and lower resolution. We compared our algorithm to uniform resolution and a simpler variable resolution algorithm; we demonstrated that our `Pruning` AMR algorithm achieves similar error tolerances to these other methods despite using many fewer degrees of freedom. However, we

also observed that our algorithm is less beneficial for INRs that are detailed throughout their entire domain, unless they are refined for many iterations. In the future, we wish to explore this direction by testing our algorithm using GPUs so that we have the capacity to run for more iterations (and thus DOFs). We also plan to test the algorithm on larger examples, such as INRs trained on full videos.

## 6 REPRODUCIBILITY STATEMENT

We will release our code if the paper is accepted for publication, but unfortunately cannot release it in an anonymized version for review. However, the algorithm presented in Section 3 gives sufficient detail to recreate our code. The software MFEM or other open source mesh refinement packages can be used to manage the AMR routines. The code provided in the original ID pruning paper (Chee et al., 2022) can be used to guide implementation of the `Prune` function in the algorithm.

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
