# OpenReview forum: "Efficient Visualization of Implicit Neural Representations via Weight Matrix Analysis"
_ICLR.cc/2025/Conference — Submitted to ICLR 2025_

### Official Review · Reviewer_YabU · 2024-10-29

**Soundness:** 1
**Presentation:** 2
**Contribution:** 2
**Rating:** 3
**Confidence:** 5

**Summary:**

The paper proposes a novel method for visualizing implicit neural representations (INRs) via an adaptive grid evaluation. The core idea is to prune the neural network using an interpolation decomposition.

**Strengths:**

* I like the presentation. It is self-contained and properly presents the background. The paper is very easy to follow.

* The core idea is simple and easy to implement.

**Weaknesses:**

* The evaluation should follow established procedures in the field. The paper uses datasets that are not usual for evaluation of similar approaches. I recommend the authors to read related papers in detail and use the datasets commonly used in the field. Example datasets include Thingi10K, Stanford, etc.

* The technical contribution is thin. The algorithm proposed may be considered incremental since the interpolative decomposition used is not proposed by the paper. In such a case, I would recommend the authors to focus on finding additional applications and to deeply evaluate the approach. That tasks would help to find additional properties of the representation that may be emphasized in future versions, increasing the manuscript value.

* There are no comparisons with state-of-the-art.

* The related works section is very thin. INRs is a gigantic area. I would advise the authors to start by checking this survey to find the papers they should cite. It is a little bit outdated now, but it is a good starting point.

```
@inproceedings{xie2022neural,
  title={Neural fields in visual computing and beyond},
  author={Xie, Yiheng and Takikawa, Towaki and Saito, Shunsuke and Litany, Or and Yan, Shiqin and Khan, Numair and Tombari, Federico and Tompkin, James and Sitzmann, Vincent and Sridhar, Srinath},
  booktitle={Computer Graphics Forum},
  volume={41},
  number={2},
  pages={641--676},
  year={2022},
  organization={Wiley Online Library}
}
```

**Questions:**

* The concept of mesh is a little bit misleading in the paper in my opinion. In the context of INRs, mesh is used to denote a surface represented by a triangle mesh. I think the correct term the paper should use is grid. That would solve other derived term problems. For example, an adaptive mesh is a concept established in Computer Graphics for decades, meaning a triangle mesh that may be subdivided or simplified as needed.

* How the coarse uniform mesh is extracted from the INR in the proposed approach?

* As algorithm 1 runs, there will be different versions of the INR matrices? Each pruning operation results in different layer matrices and the version used depends on which part of the domain is being evaluated.

* I need more details about how the pruning is applied in the algorithm in practice. $\bar{U} := UT^T$ contains the complexity that was pruned from $W$ and $b$. In other words, the pruned parameters are moved to the next layer. However, all layers should be evaluated when the INR is evaluated, thus the computation complexity still the same in the end. Probably there is an additional step to disconsider $\bar{U}$ that I did not find in the text.

* The meaning of ID_samples seems confusing. The paper first states that it is the number of samples in the domain to take when computing the interpolation decomposition (Table 1). However, in Algorithm 1 ID_samples seems to be the number of neurons to use for pruning.

* I would like to know the wall time to compute the visualization and how it compares with the non-adaptive visualization.

* Should use an usual metric to compare reconstruction (Chamfer or Hausdorff distance)

---

### Official Review · Reviewer_1F7Y · 2024-10-30

**Soundness:** 2
**Presentation:** 2
**Contribution:** 3
**Rating:** 5
**Confidence:** 3

**Summary:**

This paper addresses the challenge of efficiently visualizing implicit neural representations (INRs), which are well-suited for storing high-resolution data like time-varying volumetric data. Traditional approaches typically discretize INRs to a uniform grid, leading to a loss of the inherent advantages of INRs. To tackle this, the paper introduces an algorithm that generates an adaptive mesh by pruning the weight matrices of the INR. The key insight is that areas with low variation in the INR can tolerate more aggressive pruning than highly variable regions, enabling the mesh to be refined and adapted. This approach aims to maintain the INR's resource efficiency, even in visualization contexts.

**Strengths:**

The paper presents a compelling approach by incorporating pruning techniques into the generation of adaptive meshes based on implicit neural representations (INRs). This is an innovative idea that effectively leverages the strengths of INRs, making the visualization process more efficient and resource-conscious. The introduction of a method to visualize INRs adaptively addresses an important gap, and it highlights the potential of INRs to be used more broadly and effectively in high-resolution data applications. This direction holds promise and warrants further exploration to fully harness the benefits of INRs in visualization and beyond.

**Weaknesses:**

The paper introduces an innovative idea with significant potential, and the research direction it proposes opens exciting new avenues for leveraging INRs directly during visualization. However, despite these strengths, the paper does not feel fully 'finalized' for publication. There are several areas that would benefit from further development to strengthen its contribution. For details see below.

First, although the adaptive mesh generation from INRs is well-motivated, alternative data structures commonly used to store high-resolution data are not evaluated, and comparisons with these could provide additional insights. Additionally, the term 'visualization' may be somewhat misleading, as the method centers on adaptive mesh generation rather than actual rendering of INRs, and lacks a concrete approach for efficient visualization. Please consider defining the term "visualization" in your application more concretely.

The choice of 'Basic' as a baseline is also not well-justified, and the high-level presentation of the methodology makes it challenging to fully understand the workings of the approach. Within the paper I only found a short paragraph describing the "Basic" algorithm (l. 230-235). A more detailed description, also describing the motivation of why the authors chose this baseline would help the paper.

While the use of pruning in adaptive mesh generation is interesting, the paper could benefit from a stronger motivation for choosing pruning specifically as an optimization technique. Could the authors provide more explanation or justification for their choice of pruning as an optimization technique?

Furthermore, an analysis of storage requirements for INRs versus the adaptive mesh is missing; comparing these could provide an insightful 'upper baseline' for memory efficiency. Since INRs can be directly rendered by multiple function evaluations (albeit slowly), it would be valuable to include a performance evaluation of this approach in comparison to the proposed method, especially in the context of interactive visualization. Finally, a time-based evaluation (e.g., comparing pruning-based adaptive mesh refinement versus the Basic approach in mesh construction) would give a more comprehensive view of the method’s efficiency.

**Questions:**

- How does the proposed adaptive mesh generation from INRs compare with other data structures traditionally used for storing high-resolution data?

- Why was the "Basic" approach chosen as the baseline?

- What is the motivation for selecting pruning as the primary optimization technique, specifically for adaptive meshing of INRs?

- How do the storage requirements of INRs compare to those of the generated adaptive mesh, and could a comparative analysis be provided? And generally how do drectly visualizing the INR compare to the adaptive mesh?

- Would the authors provide a time-based evaluation comparing the efficiency of pruning-based adaptive mesh refinement against the Basic approach for mesh construction?

---

### Official Review · Reviewer_QKu8 · 2024-11-01

**Soundness:** 2
**Presentation:** 3
**Contribution:** 2
**Rating:** 3
**Confidence:** 3

**Summary:**

This paper presents an efficient method for visualizing Implicit Neural Representations (INRs) by adaptively refining only high-detail regions, identified through pruning of weight matrices. This approach maintains visualization quality with reduced memory use, as it avoids uniform discretization. Tests on CT scan data show it can achieve detailed visualization while significantly lowering computational demands, making it ideal for large-scale, dynamic data

**Strengths:**

1. The paper proposes a dynamic adaptation method for INR visualization that keeps high resolution in high-detailed region while reducing resolution in low-detailed region. This is efficient in memory saving, especially useful in large scale 3D/4D data visualization.
2. The combination of AMR and ID for variable resolution is interesting. It saves computational resources by avoiding computation on low-detailed region reconstruction.
3. It shows great potential in real-world application by the experiments in the CT dataset.

**Weaknesses:**

1. This paper lacks of theoretical analysis on how AMR and ID succeed in high-detail INR model visualization. Since AMR and ID have been widely studied and well developed. The combination is not a novel enough approach to this problem. Here are some points that I think important to analyze:
    - It is necessary to explain how important information is preserved when ID pruning in INR, especially in high-detailed regions. Since the representational capacity of INR is directly related to the size of the weight matrix, a sufficient analysis on how ID pruning affects the reconstruction accuracy of INR is important.
    - Pruning can impact the local approximation accuracy of the INR model, so it’s essential to analyze whether sufficient details can still be retained after pruning at various mesh resolutions. This aspect could be supported by a quantitative analysis on the relationship between pruning rate and error in different levels of mesh resolutions.
    - AMR relies on local error criteria, but ID pruning may reduce reconstruction accuracy in certain regions, potentially missing some details if not properly controlled. Therefore, it is necessary to analyze the impact of pruning on AMR’s local error estimation.
1. This paper gives a preliminary experiment and 2 CT experiments. The datasets are simple, not enough to support the efficiency of their algorithm. I suggest doing experiments on some medical CT datasets, e.g. [LUNA16](https://luna16.grand-challenge.org/Data/).
1. The authors show the influence of the hyperparameters to the results in their experiments, but this discussion is not enough. Across the 3 experiments, the hyperparameter $T$ varies from $10^{-4}$ to $10^{-1}$, $\epsilon$ varies from $10^{-3}$ to $10^{-2}$. The range is too big for users to find a set of useful settings. Are there any guiding rules on how to choose the hyperparameters with respect to the dataset? It's also unclear to me if the choice of $T$ and $\epsilon$ affects each other. I would suggest a more comprehensive ablation study on the choice of the hyperparameters $T$, $P$, and $\epsilon$.
1. This algorithm uses ID iteratively. I wonder if the computational cost will increase exponentially when it comes to the high dimensional dataset or large scale dataset? I suggest the authors giving a time complexity analysis with respect to the dataset scale and the dimensionality. The authors could also provide the runtime results on larger datasets (e.g. [LUNA16](https://luna16.grand-challenge.org/Data/)) if possible.

Besides, there are some minor issues:

5. There is a misspelling in the last sentence of **INPUT** in algorithm1, it should be "to" instead of "ot"
6. In algorithm1, the confition of the second for loop says M.E.done_refining == False, but I can't find anywhere that sets it false in the algorithm.
7. There are too many long sentences that take up to 3 lines. I would suggest breaking them down for reading.

**Questions:**

In this paper, authors propose a hypothesis that *the less detailed a function is on a region of the domain, the smaller an INR needs to be to accurately describe the function in that region*. Is there any verification on this hypothesis? For example, the relationship between function details and the INR size across different levels of detailed regions. The authors could provide some empirical results on the appendix. It's important to give a comprehensive verification since it is the foundation of the whole paper.

---

### Official Review · Reviewer_YBEk · 2024-11-09

**Soundness:** 2
**Presentation:** 1
**Contribution:** 1
**Rating:** 3
**Confidence:** 3

**Summary:**

The paper presents a new algorithm for visualizing implicit neural representations (INRs) through a pruning-based approach. The method determines the high-detail regions in pre-trained INRs and then uses adaptive mesh refinement to split up the domain, thus saving memory. The results show that the proposed algorithm can achieve comparable visualization accuracy while using fewer degrees of freedom than uniform grid discretization or basic AMR. However, the contribution is incremental, the presentation requires improvement, and the experimental section is weak.

**Strengths:**

1. The target problem, reducing cost in visualizing INRs, is meaningful.
2. The qualitative results show some improvement compared to AMR.

**Weaknesses:**

1. The novelty is unclear. The paper combines established techniques without adequately explaining the challenges or novel solutions provided.

2. The experiment section is weak.
The experimental results are limited; more comparisons with advanced visualization techniques for INRs would strengthen the evaluation.

3. The paper's clarity and organization could be significantly improved.

**Questions:**

1 Terminology: Key terms such as "domain" and "adaptive mesh" need clearer definitions. Are there specific examples or illustrations that could be added?

2 ID pruning

2.1 What is the computational cost of ID?

2.2 Why is the number of samples set to the width of the INR layers?

2.3 How does this hyper-parameter impact the final performance?


3 There is no detailed discussion on the computational costs of the algorithm.

4 The paper lists multiple hyperparameters but does not explain how they were chosen or their impact on the algorithm’s performance.

5 Including comparisons with state-of-the-art INR visualization methods or adaptive algorithms would deepen the insights and show the algorithm's standing in the broader research landscape.

6 Expanding experiments to larger datasets would better illustrate the scalability and robustness of the approach.

---

### Author Response · Authors · 2024-11-21
**Thank you**

Thank you to all the reviewers for their detailed comments.  We will take your suggestions into consideration as we prepare to revise this paper for submission to a different forum.

---

### Meta-Review · Area_Chair_b8rg · 2024-12-17

**Metareview:**

Though the authors did not withdraw the paper, it seems from their comment
"Thank you to all the reviewers for their detailed comments. We will take your suggestions into consideration as we prepare to revise this paper for submission to a different forum."
that they decided to withdraw the paper. In addition, the reviewers universally agree to reject the paper because of the lack of the contribution and comprehensive analysis and experiments.

**Additional Comments On Reviewer Discussion:**

There is no discussion in the rebuttal phase, so no comments on this part.

---

### Decision · Program_Chairs · 2025-01-22

Reject